# The Multifaceted Role of Curcumin in Advanced Nanocurcumin Form in the Treatment and Management of Chronic Disorders

**DOI:** 10.3390/molecules26237109

**Published:** 2021-11-24

**Authors:** Priti Tagde, Pooja Tagde, Fahadul Islam, Sandeep Tagde, Muddaser Shah, Zareen Delawar Hussain, Md. Habibur Rahman, Agnieszka Najda, Ibtesam S. Alanazi, Mousa O. Germoush, Hanan R. H. Mohamed, Mardi M. Algandaby, Mohammed Z. Nasrullah, Natalia Kot, Mohamed M. Abdel-Daim

**Affiliations:** 1Amity Institute of Pharmacy, Amity University, Noida 201303, India; 2PRISAL Foundation (Pharmaceutical Royal International Society), Bhopa l462026, India; sandeeptagde91@gmail.com; 3Practice of Medicine Department, Government Homeopathy College, Bhopa l462016, India; tagde_pooja@rediffmail.com; 4Department of Pharmacy, Faculty of Allied Health Sciences, Daffodil International University, Dhaka 1207, Bangladesh; fahadulislamdiu@gmail.com; 5Department of Botany, Abdul Wali Khan University Mardan, Mardan 23200, Pakistan; 6Integro Pharma Ltd., Dhaka 1206, Bangladesh; drzareendelawar@gmail.com; 7Department of Pharmacy, Southeast University, Banani, Dhaka 1213, Bangladesh; 8Department of Global Medical Science, Graduate School, Yonsei University, Wonju 26426, Korea; 9Department of Vegetable and Herbal Crops, University of Life Sciences in Lublin, 50A Doświadczalna Street, 20-280 Lublin, Poland; agnieszka.najda@up.lublin.pl; 10Department of Biology, Faculty of Sciences, University of Hafr Al Batin, Hafr Al Batin 39524, Saudi Arabia; esalanazy@uhb.edu.sa; 11Biology Department, College of Science, Jouf University, Sakaka P.O. Box 2014, Saudi Arabia; mogermoush@ju.edu.sa; 12Zoology Department, Faculty of Science, Cairo University, Giza 12613, Egypt; hananeeyra@gmail.com; 13Department of Biological Sciences, Faculty of Science, King Abdulaziz University, Jeddah 21589, Saudi Arabia; malgandaby@kau.edu.sa; 14Department of Pharmacology and Toxicology, Faculty of Pharmacy, King Abdulaziz University, Jeddah 21589, Saudi Arabia; mnasrullah@kau.edu.sa; 15Department of Landscape Architecture, University of Life Science in Lublin, 28 Gleboka Street, 20-612 Lublin, Poland; natalia.kot@up.lublin.pl; 16Pharmacy Program, Department of Pharmaceutical Sciences, Batterjee Medical College, Jeddah 21442, Saudi Arabia; 17Pharmacology Department, Faculty of Veterinary Medicine, Suez Canal University, Ismailia 41522, Egypt

**Keywords:** nanocurcumin, solubility, antioxidant effect, anti-inflammatory action, anticancer, neurodegenerative disease

## Abstract

Curcumin is the primary polyphenol in turmeric’s curcuminoid class. It has a wide range of therapeutic applications, such as anti-inflammatory, antioxidant, antidiabetic, hepatoprotective, antibacterial, and anticancer effects against various cancers, but has poor solubility and low bioavailability. Objective: To improve curcumin’s bioavailability, plasma concentration, and cellular permeability processes. The nanocurcumin approach over curcumin has been proven appropriate for encapsulating or loading curcumin (nanocurcumin) to increase its therapeutic potential. Conclusion: Though incorporating curcumin into nanocurcumin form may be a viable method for overcoming its intrinsic limitations, and there are reasonable concerns regarding its toxicological safety once it enters biological pathways. This review article mainly highlights the therapeutic benefits of nanocurcumin over curcumin.

## 1. Introduction

Curcumin is a bioactive compound and is the active component of *Curcuma longa* (*C. longa*), which is turmeric, a member of the ginger family. It is used as a spice, culinary coloring, and a component in ancient herbalism. Curcumin, a polyphenol, has been demonstrated to target various signaling molecules while also displaying cellular activity, contributing to its multiple health advantages. It also serves as an antioxidant, anti-inflammatory, and anticancer agent. Curcumin has been proven to have antioxidant, anti-inflammatory and anticancer effects and the ability to enhance cognitive skills and manage obesity and diabetes [1]. In Asian countries, *C. longa* has long been used as a prescription or supplement to treat diabetes, coronary disease, obesity, neurodegenerative disease, inflammatory bowel disease, allergy or asthma, and psoriasis [2]. *C. longa* is grown in tropical and subtropical climates. India is the world’s largest producer of turmeric, which has long been used as a home cure for various diseases [3]. Even though the extraction and isolation of curcumin from turmeric powder was first published in 1815, newer and more advanced extraction methods are still reported two centuries later [4]. The most frequent method for separating curcumin from turmeric has been solvent extraction followed by column chromatography, and numerous polar and nonpolar organic solvents have been utilized, including hexane, ethyl acetate, acetone, methanol, and others. For extracting curcumin, ethanol was determined to be the most preferred solvent among the organic solvents used. Although chlorinated solvents extract curcumin from turmeric quite effectively, they are not widely used in the food business because of their unacceptability. Soxhlet extraction, ultrasonic extraction, microwave extraction, zone refining, and dipping procedures have been tried, with the most being popular Soxhlet, ultrasonic, and microwave extractions [5]. Curcumin is insoluble in water; thus, it was isolated using an organic solvent. Ref. [6] developed a method for separating curcumin from turmeric powder. The authors magnetically stirred the ground turmeric in dichloromethane and heated it at reflux for one hour. The filtrate was concentrated in a hot-water bath maintained at 50 °C after being suction-filtered. Suction filtering was used to capture the reddish-yellow oil residue after it was triturated with hexane. The existence of all three components was confirmed by thin-layer chromatography (TLC) analysis (3% methanol and 97% dichloromethane) [7]. Curcumin was extracted from turmeric powder using a solvent that was a combination of ethanol and acetone. Turmeric comprises carbohydrates (96.4%), moisture (13.1%), protein (6.3%), fat (5.1%), and minerals (3.5%), according to chemical analysis. Its extracts produce curcuminoids, including curcumin (77%), demethoxycurcumin (DMC 17%), and bisdemethoxycurcumin (BDMC 3%). Curcuminoids, particularly curcumin, are used as medicines and supplements [8]. Curcumin is a symmetric molecule also known as diferuloylmethane. The IUPAC name of curcumin is 1,7-bis(4-hydroxy-3-methoxyphenyl)-1,6-heptadiene-3,5-dione, with chemical formula C_21_H_20_O_6_ and molecular weight of 368.38. Its structure has three chemical entities: two aromatic ring systems containing o-methoxy phenolic groups connected by a seven-carbon linker consisting of an α, β-unsaturated β-diketone moiety [9]. The source and chemical structure of curcumin are given in Figure 1.

## 2. Review Methodology

For this updated review, valuable data were collected from PubMed, Science Direct, Scopus, Web of Science, and Elsevier on molecular mechanism studies and pharmacological studies of natural curcumin and its nanocurcumin form. The following MeSH terms were used to search for curcumin: “chemistry”, “pharmacology”, “drug therapy”, “humans”, “clinical trials”, “diabetes”, “anti-inflammatory”, “neuroprotective”, “antibacterial”, “antifungal”, and “antiviral.” Inclusion criteria were preclinical pharmacological studies that highlighted molecular mechanisms of action and signaling pathways and experimental in vitro and in vivo pharmacological studies.

## 3. Bioavailability of Curcumin

Despite numerous health benefits, curcumin’s limited bioavailability is a fundamental criticism [10]. Low absorption, fast metabolism, chemical instability, and rapid systemic clearance have been suggested as possible causes [11]. According to various animal studies, most oral curcumin is eliminated in the feces (90%). Several strategies have been tried to boost the bioavailability of curcumin to solve this problem. Piperine, liposomal curcumin, curcumin nanoparticles, phospholipid complexes, and structural analogs of curcumin such as turmeric oil are a few examples of adjuvants [12]. Increased blood concentrations have been observed as a result of such efforts. However, human clinical trials comparing the therapeutic potencies and pharmacodynamic responses of these more bioavailable variants to those of conventional curcumin have yet to be undertaken extensively. Furthermore, the serum concentrations required to achieve a particular clinical or biological effect have yet to be determined. Piperine is an alkaloid found mainly in *Piper nigrum*, and it is this component provides black pepper its piquancy. Piperine has been shown to increase curcumin bioavailability [12]. Piperine suppresses the liver enzyme UDP-glucuronyl transferase, reducing the amount of glucuronidation of curcumin. Additionally, curcumin is accessible for ingestion as a result of this mechanism [13]. An oral dosage of 2 g/kg curcumin with 20 mg/kg piperine was delivered concurrently to animals and mice in an in vivo investigation [14]. Curcumin’s relative bioavailability was enhanced 1.54-fold in rats and 20-fold in healthy volunteers. Even though the increase in curcumin bioavailability was higher in humans than in rats, the quantity of curcumin ingested was more remarkable in rats than in humans. In another clinical experiment, healthy human volunteers were given 2 g of curcumin and 5 mg of piperine daily. The research found that administering curcumin plus piperine together boosted absorption by 200% then consuming curcumin alone [15]. Zeng et al. [16] looked at the effect of piperinepre administration on curcumin oral bioavailability and in this investigation, rats were given 20 mg/kg piperine first and subsequently 200 mg/kg curcumin at frequencies of 0.5 to 8 h after piperine treatment. Compared to those that received pure curcumin, the rats that received piperine before the curcumin exhibited a statistically huge increase in curcumin oral bioavailability, notably at 6 h following ingestion of piperine, with AUC0-t increasing 97-fold [17]. Thus, based on the aforementioned studies, it has been found that delivering natural substances such as piperine, quercetin, resveratrol, and silibinin in combination with curcumin resulted in improved curcumin absorption. This is a potentially cost-effective way to increase curcumin’s oral bioavailability and further explore it in developing novel drug delivery systems [18].

Nanocarrier-based delivery of curcumins one of the best approaches to improve curcumin’s solubility and bioavailability while also protecting it from hydrolysis-induced inactivation. Some nanocarriers emphasized long-term retention and circulation in the body, while others focused on intracellular release mechanisms and cellular delivery. Curcumin is solubilized in these environments by becoming entrapped in hydrophobic pockets, primarily through hydrophobic interactions. Curcumin’s fluorescence is boosted when solubilized in any of these systems, making it simple to measure its binding effectiveness. Because of their biocompatibility, these systems may be successfully studied for anticancer activity in cancer cells and in vivo systems, with considerable increases in anticancer activity due to enhanced curcumin bioavailability reported. Curcumin liposomal formulations have been proven to be the most effective for increasing curcumin bioavailability in cells [19], and products based on liposomal formulations are being commercialized. Nanocurcumin’s effectiveness is due to the size, surface area, charge, and hydrophobic nature of the particles, which make it superior to native curcumin [20] and regarded as an acceptable target for usage as a drug compared to standard curcumin. This feature is especially crucial in the fight against infectious illnesses caused by intracellular infections [21]. According to Ma et al. [22], nanocurcumin increased in vivo bioavailability and tissue distribution, with a 60-fold increase in biological half-life compared to native curcumin treatment in rat models.

## 4. Therapeutic Applications of Nanocurcumin over Curcumin

Nanocurcumin has the potential to prevent and treat a wide range of human diseases. Successful therapeutic applications of nanocurcumin are highlighted in the following section.

### 4.1. Antioxidant Effects

The antioxidant property of curcumin is due to its structure, which interconnects two methoxylated phenols with two unsaturated carbonyl groups grouped in stable enol form [23]. The phenolic O-H and C-H groups in curcumin are responsible for its bioactivity. Curcumin also plays a role in lipid peroxidation inhibition by oxidizing a polyunsaturated fatty acid called linoleate, which is oxidized to generate a fatty acid radical. Curcumin also participates in the intermolecular Diels–Alder reaction by breaking the chain at the 3’ site and neutralizing lipid radicals [24]. Besides lipid peroxidation inhibition, curcumin has been shown to have free radical scavenging activities in in vitro and in vivo models employing rat peritoneal macrophages. Curcumin actively scavenges numerous reactive oxygen species (ROS) produced by macrophages, such as hydrogen peroxide, nitrite radicals, and superoxide anions [25].

Nanocurcumin therapy is a promising method in modulating inflammatory cytokines, particularly IL-1 and IL-6 mRNA expression and cytokine production in COVID-19 patients, resulting in a better clinical presentation and over all recovery [26]. Significant levels of nitric oxide (NO) were created by iNOS, which then reacted with superoxide radicals in the surrounding oxidative environment to make peroxynitrite, a poisonous compound that was proven to be fatal to cells. Curcumin has been shown to inhibit the activity of iNOS and reduce ROS levels in cells. Additional research on microglial cells has demonstrated that this antioxidant spice can protect brain cells from oxidative damage by lowering NO production and diminishing neuroinflammation associated with chronic neurodegenerative diseases such as AD [27]. Alg-NP-Cur was analyzed and evaluated in a PD model in drosophila. It exhibited good antioxidant activity by decreasing lipid peroxidation in the PD drosophila brain after a diet supplemented with the nanocarrier for 24 days [28]. Curcumin nanocrystals were used to protect Wistar rats from cardiovascular damage by lowering lipid peroxidation and enriching the properties of antioxidants [29]. Using an aluminum phosphide (AIP) toxicity-induced rat model, Ranjbar et al. [30] studied the effects of curcumin and nanocurcumin on the oxidant and antioxidant system in the liver mitochondria and found that nanocurcumin enhanced oxidative stress variables and protected the liver by scavenging free radicals and stabilizing the oxidative state.

### 4.2. Anti-Inflammatory Effects

Curcumin aids in the battle against invaders and aids in the healing of the rift zone. While acute, short-term inflammation is helpful, it may become a problem if it becomes persistent and destroys cells. Excessive inflammatory activation for long periods of time can result in mitochondrial dysfunction. Traumatic brain injuries (TBI), spinal cord injuries (SCI), and hemorrhagic/ischemic stroke all cause substantial changes in mitochondrial dynamics, notably increased membrane permeabilization, oxidative phosphorylation, and the accumulation of mitochondrial ROS [31,32]. Excessive glial activation has been linked to the same outcomes as long-term inflammation. Advanced mitochondrial dysfunction has also been demonstrated to increase inflammatory processes, leading in neuronal injury and poor neurological consequences [33,34]. Studies have identified that the gene CDGSH iron sulfur domain 2(CISD2) protects against inflammatory reactions and apoptosis caused by mitochondrial dysfunction. The presence of CISD2 in the outer membrane of mitochondria has also been linked to the preservation of mitochondrial integrity. CISD2 deficiency resulted in mitochondrial dysfunction and cell death, according to reports on CISD2 knockout mice [35,36]. The attachment of BCL2 to BECN1, which governs cellular autophagy/apoptosis, has been found to be aided by CISD2. Anti-inflammatory and/or antiapoptotic therapeutics based on CISD2 are extremely likely to be developed to combat the consequences of aging, neurodegenerative disease, and CNS trauma [37].

A variety of radical oxygen/nitrogen species may also trigger a signal transduction pathway that boosts proinflammatory neural activity. Inflammation has been linked to the onset of a variety of chronic illnesses and disorders, as illustrated in Figure 2: Alzheimer’s disease (AD), Parkinson’s disease (PD), multiple sclerosis, epilepsy, brain damage, colorectal cancer, metabolic disorders, carcinoma, allergies, asthmatic, pneumonia, diarrhea, osteoarthritis, renal ischemic, lupus, diabetes, overweight status, anxiety, tiredness, and acquired immunodeficiency syndrome (AIDS) [38].

The stimulation of a transcriptional nuclear factor kappa B (NF-κB) regulates the impact of tumor necrosis factor (TNF-), a valuable inflammatory marker in most illnesses. Even though TNF- is the most potent NF-κB activator, TNF- production is low. NF-κB is also known to regulate this gene. TNF- is not the only cytokine that can be found in most cancers. Gram-negative bacteria; numerous disease-causing viruses; air toxins; biological, bodily, structural, and mental anxiety; high glucose; fatty acids; UV exposure; tobacco smoke; and other illness factors are also known to activate NF-κB [39]. As a result, medicines that inhibit NF-κB and NF-B-regulated gene products may be effective against various illnesses. Turmeric has been found to inhibit NF-κB activation, which is induced by several inflammation stimuli. Turmeric has also been demonstrated to reduce inflammation via different methods outside the scope of this study, indicating that it may be used as an anti-inflammation drug [40].

Wang et al. [41] produced curcuminsolid lipid nanoparticles (SLNPs) and improved efficacy in an allergic rat model of ovalbumin-induced asthma. According to Milano et al. [42], nanocurcuminis efficient against the Ehrlich ascites carcinoma (EAC) cell lines OE33 and OE19. It makes EAC cells more vulnerable to T cell-induced cytotoxicity and reduces T cell proinflammatory signals. Nanocurcumin improves oral bioavailability and, therefore, efficacy in preventing streptozotocin (STZ)-induced diabetes in rats, at least in part, by suppressing inflammation and pancreatic beta-cell death as compared to the native form. According to another investigation, loss of NF- activation led to downregulation of COX-2 and iNOS expression, blocking the inflammatory response and carcinogenesis [43]. W.S. Sarawi et al. [44] examined the impact of curcumin and nanocurcumin on copper sulphate (CuSO_4_)-induced brain oxidative stress, inflammation, and apoptosis in rats, and found that Akt/GSK-Akt/GSK-3β may be involved. Rats were given 100 mg/kg CuSO4 and were also given curcumin and nanocurcumin for 7 days. Cu-administered rats had significantly higher levels of MDA, NF-κB p65, TNF-, and IL-6 in the brain, as well as lower levels of GSH, SOD, and catalase. Cu produced DNA fragmentation in rats’ brains, elevated BAX, caspase-3, and p53, and decreased BCL-2, MDA, and NF-B p65. Nanocurcumin and curcumin, on the other hand, lowered proinflammatory cytokines while also decreasing proapoptotic genes, increasing BCL-2, and improving antioxidants and DNA integrity. In the brains of Cu-administered rats, both nanocurcumin and curcumin enhanced AKT Ser473 and GSK-3β Ser9phosphorylation. Finally, nanocurcumin and curcumin protected against Cu neurotoxicity by reducing oxidative damage, inflammation, and apoptosis while also increasing AKT/GSK-3AKT/GSK-3βsignaling. Nanocurcumin had a stronger neuroprotective impact than curcumin. Sinjari et al. [45] also investigated the anti-inflammatory action of curcumin liposomal formulations (CUR-LIP) and found it to be potent.

### 4.3. Antibacterial Effects

The dramatic rise in microbe resistance to multiple medications has necessitate daquest for novel and potentially effective antibacterial agents with minimal or lower human cytotoxic effects that can aid in treating a wide range of microbial diseases [46]. Curcumin antibacterial effectiveness against Gram-positive bacteria (*Staphylococcus aureus*) and Gram-negative bacteria (*Escherichia coli*) was studied by Tyagi et al. [47]. Curcumin was shown to have good antibacterial activity against all tested microorganisms [48]. Curcumin’s increased antibacterial action in the form of sunlight radiation mainly owed to photoexcitation, which caused the production of ROS that inhibited bacteria load [48].

Bhawana et al. [49] utilized the wet milling process to create water-soluble nanocurcuminin the 2–40 nm range of sizes and tested its antibacterial properties. *S. aureus*, *B. subtilis*, *E. coli*, *P. aeruginosa*, *Penicillium notatum*, and *Aspergillus niger* were shown to be resistant to nanocurcumin. Bhawana et al. also found that gram-positive bacteria had more potent antibacterial activity than Gram-negative bacteria. Similarly, employing surface-charged sodium dodecyl sulfate, Tween 20, and cetrimonium bromide, No et al. [50] developed nanocurcumin via an acoustoplastic deformation method. The investigators used *L. monocytogenes*, a gastrointestinal parasite, to evaluate the nanocurcumin they created. Shariati et al. [51] recently synthesized N-CUR and studied it against multidrug-resistant *P. aeruginosa*. The researchers observed that 128 g/mL of soluble curcuminin dimethyl sulfoxide (DMSO) inhibited bacterial growth, but 256 g/mL was necessary for soluble curcumin in DMSO to suppress it. At 64 g/mL, the nanocurcumin inhibited biofilm development and destroyed 58% of the biofilm. Based on these findings, the researchers hypothesized that nanocurcumin might lower *P. aeruginosa* pathogenicity and biofilm formation. Polymeric liposomes, micelles, and nanoparticles have all been described as intriguing biological systems. Nanocurcumin is quite desirable and effective for the treatment of different illnesses because of its excellent loading capacity and diminutive size. Curcumin therapeutic efficacy is increased when combined with nanomaterials, according to several kinds of research [52]. The antibacterial activity of silver-decorated polymeric micelles coupled with curcumin was investigated by Huang et al. [53].

### 4.4. Hepatoprotective Effects

Curcumin nanoliposomes were developed to lower the particle size while simultaneously increasing hydrophilicity [54]. The hydrodynamic layer around the nanoparticles was smaller as a consequence, resulting in a faster rate of surface-specific dissolving. It was said that liver damage, including necrosis and steatosis, might lead to a short event of the hepatotoxic substance carbon tetrachloride (CCl_4_). When the liver was exposed to untreated curcumin or curcumin nanoliposomes, the blood activities of alanine transaminase (ALT), aspartate aminotransferase (AST), and alkaline phosphatase (ALP) were significantly decreased. It was also discovered that the study group given curcumin nanoliposomes had less ALT, AST, and ALP serum activity than the study group provided untreated curcumin. This demonstrated that curcumin nanoliposomes might function as the most effective medication for treating any liver injury and might have a successful hepatoprotective impact [54]. Nanocurcumin was utilized for about the first time in liver tissues to manage the deleterious effects of CuSO_4_, since wat is thought that nanocurcumin’s compact size might be handy for associating with outer and inner biomolecules effectively [55]. Maghsoumi et al. [56] conducted research primarily to assess the preventive and rejuvenating properties of curcumin nanomicelles on prolonged liver damage produced by alcohol in mice. The findings indicated that nanocurcumin had a substantial effect on preventing the liver from CCl_4_-induced damage, and it was also found that nanocurcumin’s hepatoprotective activity was based on a specific dosage of 2 mL/kg of body weight [56]. A further study [57] found that in obese individuals with nonalcoholic fatty liver disease, nanocurcumin increased high-density lipoprotein (HDL) levels and lowered fatty liver and liver transaminase levels, demonstrating that nanocurcumin had a hepatoprotective effect.

### 4.5. Lower Risk of Heart Disease

Cardiac issues are the leading cause of mortality worldwide. Researchers have studied it for years and have learned a great deal about why it occurs. This chronic problem is unexpectedly complex, and various factors play a role in its development. Curcumin’s primary advantage in cardiovascular disease (CVD) is that it improves the performance of the endothelial, which lines blood vessels [58]. Curcumin has been linked to improved heart health in many studies [58]. Curcumin promotes cardiac repair and ameliorates cardiac dysfunction following myocardial infarction [59].

Although curcumin has been proven to positively impact on CVD and its risk factors, however it has a limited effectiveness due to its poor aqueous solubility. After oral dosing, this is ascribed to absorption, increased bioavailability, and systemic clearance. As a result, its therapeutic effects are substantially reduced [60]. Even though curcumin and nanocurcumin have the same crystal structure, nanocurcumin can be as helpful as curcumin in reducing CVD risk factors. Earlier in vitro and in vivo investigations, for particular, have shown that micro may be better than native curcumin in terms of certain testing products [61].

### 4.6. Nanoencapsulated Curcumin’s Potential in Crossing the Blood–Brain Barrier (BBB)

#### 4.6.1. Boost Brain-Derived Neurotrophic Factor

Nerve cells can establish social contacts, and they may increase and grow in quantity in specific regions of the brain. Brain-derived neurotrophic factor (BDNF) is one of the most significant factors in this process [62]. It is a gene that is engaged in the manufacture of an enzyme that helps neurons live longer. The BDNF protein is located in regions of the brain that control feeding, drinks, and fat mass, and it plays a role in cognitive function [63]. Low levels of BDNF protein have been related to various common brain diseases, including stress and AD. Curcumin has been shown in animal experiments to boost BDNF levels in the brain [64]. It may be possible to postpone or even reverse many brain illnesses and age-related declines in brain function [65].

When primary tumors or metastases develop in unhealthy circumstances, the BBB is physically and functionally impaired beyond 1–2 mm in diameter, during this stage it becomes the blood–brain tumor barrier (BBTB). The deficiency of fenestration, the tight connections between endothelial cells (ECs), and the lower incidence of pinocytosis all restrict molecular exchange across the BBB, limiting transcellular transit [66]. Curcumin’s limited penetration through the BBB and its chemical–physical properties provide significant constraints for using these novel systems in brain diseases. The BBB between the blood and the central nervous system (CNS) has the physiological function of maintaining ionic homeostasis and chemical exchange, thereby shielding the CNS from infections and circulating toxins and sustaining the brain microenvironment [67]. The structure, function, and permeability of the BBB changed in brain cancers such as glioblastoma (GBM). Even though various methods for crossing the BBB have been developed, significant issues still make brain drug delivery problematic. The use of customized nanocarriers to preserve drugs while allowing them to pass across the BBB appears to be a successful approach [68].

The first stage of endocytosis determines the ultimate passage of nanoparticles into endothelial cells via transcytosis. The endothelial cells’ outer membrane influences the nanoparticles’ entrance; in fact, positively charged nanoparticles employ the adsorbing transcytosis route more readily than neutral or negatively charged nanoparticles, which have more negligible protein adsorption and thus longer circulation durations. Because of improper tight junctions, nanoparticles can penetrate the BBB via receptor-mediated transcytosis (RMT), adsorptive-mediated transcytosis (AMT), and extended permeability and retention (EPR) [68].

Endothelial cells (ECs), pericytes, and astrocytes occupy the BBB, and their end feet cover the basal lamina of the brain capillaries. Small soluble substances and endogenous macromolecules pass through the BBB via various transport pathways classified into diffusion and transcytosis (Figure 3). Hydrophilic and lipophilic compounds diffuse across cells via tight junctions or migrate through cells by dissolving in the plasma membrane [68]. Several aspects must be evaluated for proper drug transport across the BBB utilizing nanofabricated devices, including size, charge, biocompatibility, blood circulation stability, and controlled release capacity. Nanoparticles have been shown to penetrate the BBB efficiently [69]. Nanoparticles with diameters of 50 to 100 nm are most effective in animal models of CNS disorders. The specificity of BBB nanosystems, typically established by functionalizing them with particular proteins, is also critical [70].

#### 4.6.2. Potential Role in the Treatment of Neurodegenerative Disease

Despite the limited number of in vivo, in vitro, and clinical investigations, data on the bioactive function of various nanoencapsulated curcumins in protection against and control of many CNS-related diseases have suggested nanocurcumin as a new approach to neurodegenerative diseases, including AD and PD [68]. Meng et al. [71] explored the effectiveness of a novel nanostructured low-density lipoprotein transporter that was tailored with curcumin-loaded lactoferrin for a targeted brain release that might potentially decrease the progression of AD. This nanocurcumin formulation, with a size of 100 nm, penetrated the BBB and released curcumin efficiently. The nanostructure had a protective impact on neuronal injury, according to histopathological evaluation. Furthermore, plasma levels of malondialdehyde (MDA), a key biomarker of lipid peroxidation, were lower in the nanocurcumin-treated groups, indicating BBB bridging and efficacy in decreasing oxidative damage [72]. It is well known that oxidative stress is linked to NO generation and mitochondrial dysfunction in the pathophysiology of PD [73]. Curcumin-loaded solid lipid nanoparticles (SLCN) were efficiently developed to have a particle size of about 86 nm and show no toxicity in endothelial brain cells. The effectiveness of SLCN against lipopolysaccharide-induced neuroinflammation in BV-2 microglial cells was also investigated. In a dose-dependent way, the SLCN inhibited NO generation more effectively than conventional curcumin [74].

##### Potential Role in AD

Curcumin has been proven to penetrate the venous barrier, so there may be some excellent signs on the way. Inflammatory and lipid peroxidation are known to have a role in AD. An accumulation of protein tangles known as an amyloid plaque is also a major characteristic of AD. Curcumin has been shown in studies to aid in the removal of these plaques. Curcumin’s ability to delay or perhaps reverse the development of AD in humans is unclear at this time, and more research is needed on this topic [75].The inclusion of curcumin into cultured neural stem cells has been shown to reduce brain risk of certain types of β-amyloid-induced neurotoxicity. When β-amyloid is used to create animal models of AD, curcumin guards against damage [76].

Curcumin prevents the agglomeration of β-amyloid into folded layers and reduces the production of β-amyloid from isozyme. Turmeric also has a preventive impact on mouse models against overproducing β-amyloid, according to studies. In these animal studies, brain activity was also enhanced. Curcumin clinical studies in AD have not proven particularly encouraging. This is likely due to turmeric’s low biocompatibility in humans; therefore, researchers have been working on ways to enhance delivery methods or produce analogues that replicate the neuroprotective benefits while being readily absorbed by the brain [77]. Curcumin’s lack of effectiveness in people with AD may possibly be due to treatment for too little time or beginning therapy too late in the illness’s progression, when significant neuronal death has already happened and cannot be reversed. Curcumin may be helpful in preventing the formation or advancement of AD. Many in vitro and in vivo investigations have examined how turmeric interacts with β-amyloid. Curcumin’s dose-dependent preventive activity against β-amyloid-induced damage in established neural stem cells has been widely studied. Several factors have been described to explain this preventive role. Inhibiting NF-κB with curcumin prevented β-amyloid-induced cell death in both human monocytes. Curcumin also inhibited NF-κB-induced suppression of peroxiredoxin-6 in rat hippocampus cells, reducing hypoxia-induced cell death [77].

Curcumin was found to reduce β-amyloid-induced affectation of the cytokines tumor necrosis factor-α and interleukin-1, as well as stimulation of mitochondrial protein kinase and phosphatases of intracellularly message kinase-1/2, in a human acute monocytic leukemia cell line (Sigma-Aldrich) [78]. The adding of turmeric to rat prefrontal cortical neurons reduced the β-amyloid-mediated rise in caspase-3 and triggered the defensive pathway involving Akt. Curcumin preserved cell viability in rat cortical cells following exposure to β-amyloid and reduced oxidative stress indicators and reactive oxygen species levels. Curcumin also seemed to decrease β-amyloid toxicity in APPs transfected SY5Y cells by reducing GSK-3 activity and boosting the protective Wnt/β-catenin pathway. As a result, curcumin seems to work on many levels to reduce tissue damage induced by inflammatory, peroxidation, or β-amyloid exposure [78].

Curcumin may also influence the formation and storage of β-amyloid, a protein that has long been believed to be a cause of neurotoxicity in AD [79].

Curcumin may be beneficial by lowering maintenance levels, according to other studies. Turmeric reduced the amount of amyloid precursor protein (APP) and prevented the formation of β-amyloid 1–42 in tissue culture. Curcumin did not seem to alter APP degradation by β-site amyloid precursor protein cleaving enzyme 1 (BACE-1), an enzyme implicated in the formation of –β-amyloid. Neither BACE-1 protein nor messenger planning to open acid levels was changed; thus, the action must involve an APP signal transduction process. Curcumin did not affect the amounts of mature APP in a later study, although it did reduce the levels of immature and total APP. The authors of [80] hypothesized that by disrupting the route that leads to the formation of both β-amyloid 1–40 and β-amyloid 1–42, inhibiting the APP proliferative phase may explain the participative management of both β-amyloid 1–40 and β-amyloid 1–42 [80]. Curcumin seems to influence the action of β-secretase via lowering the production of closed air, the enzyme’s active subunit. This may be due to glycogen synthase kinase-3 (GSK-3) suppression, which hydrolyzes presenilin-1 to activate β-secretase. GSK-3 activity may hydrolyze tau, enabling it to form Hrithik filaments in order to raise β-amyloid levels.

Curcumin’s capacity to bind iron is another method by which it may reduce the development of β-amyloid. Increases in iron may promote the creation of β-amyloid. Curcumin reduces iron-induced toxicity in basic cultures, and it has been suggested that turmeric’s therapeutic benefits in nova animal models of AD may be attributable in part to its iron-related actions. Curcumin may reduce β-amyloid formation and depositing via a variety of pathways, including changes in secretase activity and amyloid precursor protein (APP) maturity, decreasing GSK-3 activity and tau synthesis, and sorption of iron. Table 1 summarizes these activities [81].

Curcumin nanoformulation by altering the surface of the PLGA polymer and encapsulation of selenium nanoparticles decreased the Aβ load and inflammation in brain samples of AD mice and treated the memory loss of the model mice. Moreover, histopathological images of animal tissues displayed no deceptive abnormalities [82]. Singh et al. [83] developed curcumin-encapsulated Pluronic F127 nanoparticles (FCur-NPs) and compared the blood–brain barrier penetration of free curcumin (free CUR) and FCur-NPs. In the brains of mice, FCur-NPs had a 6.5-fold higher fluoresce intensity than free CUR. In vitro comparisons with Congo red then demonstrated that encapsulated CUR retains its capacity to bind to A plaques. When compared to free CUR, FCur-NPs showed antioxidant and antiapoptotic activities. In vitro and in vivo data implied that FCur-NPs might be useful as a theragnostic agent for AD.

##### Potential in the Treatment of GBM

Curcumin has been used as an adjuvant to GBM therapies in a number of studies. Various experimental models of GBM treated with different curcumin nanoencapsulations (methoxy polyethylene glycol polycaprolactone, mPEG-PCL; solid lipid nanoparticles, SLCP; curcumin-loaded lipid-core nanocapsules, C-LNCs) showed similar results, such as efficient endocytosis, increased apoptosis, and arrest of the G2/M cell cycle [84]. Curcumin encapsulated in nano micelles inhibited U373 cell growth by modulating the NF-κB pathways, resulting in early G2/M cell cycle arrest, increased sub-G1 cell cycle arrest, and cell damage induction [80]. A further new dendrosomal curcumin (DNC) prohibited U87MG cell proliferation in a time- and dose-dependent manner, increasing the number of apoptotic cells when used in combination with p53 overexpression [85]. In another study, curcumin-loaded zein nanoparticles (CUR-ZpD-G23 NPs) successfully crossed the BBB and administered curcumin to glioblastoma cells. When compared to free curcumin, the NPs increased cellular absorption of curcumin by C6 glioma cells and demonstrated great penetration into 3D tumor spheroids. BBB crossing and tumor spheroid penetration were increased by G23-functionalized NPs. Furthermore, in liquid and soft agar models of C6 glioma cell development, the NPs significantly reduced proliferation and migration, as well as causing cell death [86]. Finally, curcumin nanoencapsulation in polysaccharide matrices (diameters 210–240 nm) based on hyaluronic acid (HA) and curcumin–lactoferrin conjugated (Lf-Cur-PSNPs) showed excellent BBB penetration. Lf-Cur-PSNPs were occupied predominantly by brain capillary endothelial cells and remained constant and more efficient in targeting C6 glioma cells after crossing the BBB [87].

### 4.7. Anticancer Effects

Curcumin is efficacious against cancer by inhibiting all three phases of tumorigenesis: start, development, and advancement [88]. Curcumin is a chemosensitizer and radio-sensitizer for tumor cells, and a chemoprotector and radioprotector for body cells, in addition to its chemopreventive ability with low discomfort and susceptibility [89]. Curcumin is a high-epigenetic-regulation-potential anticancer drug that acts with many cancer targets and processes. It can prevent tumorigenesis through various mechanisms, including tumor apoptosis, inhibition of proliferation, antiangiogenesis, and stimulation of mitotic catastrophe, differentiation and atrophy, suppression of cytokines and migration, and genome control [90].

#### 4.7.1. Activation of Tumor Apoptosis

In many cancer forms, curcumin therapy may trigger apoptosis (the process of programmed cell death; opposition to apoptosis can encourage tumor formation) via internal stress responses and external ligands. Cytotoxicity is initiated intrinsically in response to cellular cues such as stress and DNA damage. Turmeric was found to have two purposes: it upregulates proapoptotic B-cell lymphoma 2 (Bcl-2) family enzymes (Bim, Bax, Bak, Puma, and Noxa) and downregulates the antiapoptotic proteins X-linked inhibitor of apoptosis protein (XIAP), Bcl-2, and B-cell lymphoma-extra-large (Bcl-xL) [91]. The NF-κB regulated discovered genes Bcl-2, Bcl-XL, cIAP, survivin, and tumor necrosis factor receptor (TNFR) associated factors 1(TRAF1) and 2 (TRAF2), as well ascurcumin, were also shown to operate mainly via inhibiting the apoptotic pathway [92]. Autophagy can be triggered at the binding site via the intrinsic route, which fuels the growth of cellular membranes transmitters (Fas, TRAIL). Unlike the inner pathway, which is linked to endorphins, the extrinsic pathway is linked to death receptors. This route leads to the formation of the Fas, flavin adenine dinucleotide (FAD), and caspase-8 and -10 death-inducing signaling complexes. The activation of cell damage by apoptosis after combination therapy lends credence to curcumin’s potential effectiveness as cancer care. Another immune cell enzyme, p53, may be activated by curcumin and mediates apoptosis death in basal pituitary tumor lines in a dose- and time-dependent manner [93]. The mitogen-activated protein kinase (MAPK) family is a large set of cellular proteins known to bind p53’s trans activation domain (c-Jun N-terminal kinases (JNK) 1-3, ERK1/2, and p38 MAPK). On the other hand, curcumin has been shown to promote apoptosis by phosphorylating and activating JNK [94].

#### 4.7.2. Antiangiogenesis

Curcumin stimulates neurogenesis (the process of supplying food and energy to a tumor while also removing toxins) by controlling endothelial features, which include a few other metabolic enzymes, e.g., basic fibroblast growth factor (bFGF), epidermal growth factor (EGF), granulocyte colony-stimulating factor, interleukin 8(IL-8), platelet-derived growth factor(PDGF), transforming growth factor–tumor necrosis factor (TGF– TNF), vascular endothelial growth factor (VEGF),and many small particles (e.g., adenosine, prostaglandin E). VEGF and bFGF seem to be the most significant of these chemicals in maintaining tumor development [95]

Attacking VEGF or its cellular receptor is a common strategy for anti-VEGF treatment. Antigens are the most frequent neurotransmitter medicines, while receptor tyrosine kinase agents are the most common direct amplifiers of VEGF activity. Curcumin inhibited VEGF and angiopoietin 1 and 2 gene production in EAT cells, VEGF and angiopoietin one oxidative stress in NIH3T3 cells, and KDR gene expression in HUVEC cells in a time-dependent manner [96]. Curcumin shown to inhibit VEGF production in blood clotting via mechanisms such as TGF release, COX-2 hypertrophy, hydrogen peroxide discharge, conventional and aberrant EGFR, Src signaling, and abnormal NF-B signaling [97]. In mice, curcumin and its derivatives inhibited primary b-FGF-mediated ocular neovascularization significantly. Curcumin has also been shown to decrease VEGF and b-FGF biomass production in estrogen receptor-negative MDA-MB-231 tumor tissues [98].

Vasculature, epithelial cell movement, and tube construction are all mediated by matrix metalloproteinases (MMPs). Gelatinase A and gelatinase B are metalloproteinases that activate transcription factors to develop new microvascular. Curcumin has been found to inhibit 72kDa MMP production and transposition. Phytochemicals suppress the production of gelatinase B, which is activated by the FGF-2-regulated gene transcription activator protein 1(AP-1) [99]. CD13/aminopeptidase N (APN) is a zinc-dependent vesicle protease involved in tumor invasion and morphogenesis. Curcumin linked to APN and reduced its function irreversibly, according to Shim and colleagues [100]. Liposomal curcumin reduced tumor size and decreased CD31 production, as well as VEGF and IL-8 inflexion, suggesting that curcumin prevented tumor angiogenesis and lowered pancreatic cancer development in mouse xenograft models [101]. Curcumin also had the capacity to incite apoptosis, inhibit cancer cell proliferation, and decrease cell cycle development, making it a potential therapy against human lung, breast, prostate, pancreatic, and melanoma malignancies [102]. Some researchers have claimed that further clinical trials on curcumin are unnecessary because of its fragile, reactive, and non-bioavailable nature [103]. Curcumin has been found to suppress cancer cells from growing. Curcumin inhibits the function of matrix metalloproteinases, which govern the process, preventing cancer cells from attacking normal tissue. Curcumin suppresses the production of genes including cyclin D1, c-myc, bcl-2, and Bcl-xL, which are implicated in tumor growth, proliferation, and apoptosis. NF-κB inhibition, for example, is crucial in carcinogenesis and proliferation [104]. Basniwal et al. [105] investigated the anticancer effects of curcumin nanoparticles in cancer cell lines from the lungs (A549) and skin (A431). In aqueous circumstances, curcumin nanoparticles were shown to have a significantly greater effect on cancer cells than inherent curcumin. One of the most common histological subtypes of breast cancer with a metastatic characteristic is triple-negative breast cancer (TNBC). Dendrosomal nanocurcumin and exogenous p53 have had anticancer effects on TNBC cells when used together [106].

### 4.8. Antidiabetic Effects

Diabetes and heart disease model mice (db mice) received dietary curcumin (0.2 g/kg, 6 weeks) that lowered HOMA-IR and HbA1c levels and the activity of the hepatic gluconeogenic enzyme. Curcumin raises the blood levels of adiposity, an adipocytokine that improves insulin resistance [107]. A few findings on curcumin’s antiobesity and antidiabetes benefits are included here. Ina study in which 14 healthy individuals were given 6 g turmeric powder along with glucose and observed no impact of oral curcumin consumption (capsules) on glucose tolerance. However, they discovered an increase in blood insulin. Another scholar performed a randomized quintuple placebo-controlled study in 240 people with the post as an example. For nine months, a group of patients was administered a curcumin capsule containing turmeric extract (75–85% curcuminoids) at a dose of 1.5 g per day. Insulin secretion improved considerably, and blood triglyceride levels increased, compared to the control group [108]. This same study team conducted a 6month experiment with 213 people with diabetes at risk of angiogenesis, in which one group of subjects received identical pills as in the prior trial. They had lower HOMA-IR, reduced central body fat, and more significant plasma adiponectin64 than the placebo group. An oral intake of 300 mg/day glucosamine composition (curcumin 36.06%, dimethoxycurcumin 18.85%, bisdemethoxycurcumin 42.58%) for 12 weeks resulted in lower HbA1c and HOMA-IR levels compared to the placebo group in a randomized, double-blind placebo clinical study consisting of a total of 100 people with diabetes with fatness. Several good reviews [109,110] may aid in gaining a better knowledge of the antidiabetes and antiobesity benefits. In rats and mice, foliar spray enriched diets substantially reduced body fat accumulation and improved hyperglycemia. Instead of turmeric, large dosages of ferulic acid were given in each instance. According to human research examining different polyphenol metabolites, ferulic acid secreted in urine is not substantially linked with type 2 diabetes risks. There have been no reports on vanillin, dicyclopentadiene, or curcumin glucuronic acid, curcumin’s major precursor [111]. Nanocurcumin has been shown to be efficacious against diabetes in rats caused by STZ in research. Nanocurcumin mitigated oxidative stress and lowered inflammation and apoptosis in pancreatic β-cells [112]. Curcumin promoted glucose uptake in cells by increasing GLUT4 translocation and enhancing insulin levels in body cells, according to Mohiti et al. [113].

## 5. Various Nano Drug Delivery Systems for Curcumin

### 5.1. Liposomes

Natural phospholipids can be used to produce liposomes, which are globular synthetic vesicles. They are inner enclosed colloidal materials made up of lipid bilayers with an exterior lipid bilayer around a core aquatic region [114,115]. Liposomes range in size from 25 to 205 nm in diameter. Liposomes are supposed to be utilized as drug transporters and immunological assistance agents and are encapsulated either in the aquatic part of the lipid bilayers or at the bilayer surface. They can be used to encapsulate drugs with a wide range of solubilities or lipophilicities [116,117]. Furthermore, they can carry drugs into cells by fusion or endocytosis, and almost any chemical, regardless of solubility, can be encapsulated in liposomes (Figure 2). To enhance curcumin solubility, Rahman et al. [118] produced β-cyclodextrincurcumin binding interactions that incorporated both natural curcumin and the complexes independently into liposomes. Entire curcumin-containing preparations were efficient at suppressing cell growth in vitro cell study [119]. Shi et al. [120] used an enzyme-linked immunosorbent assay (ELISA) technique to evaluate curcumin’s therapeutic benefits on lung fibrosis in mice using a water-soluble liposomal curcumin that was found to successfully reduce radiation pneumonitis and lung fibrosis and sensitize LL/2 cells to irradiation. Some research has indicated that liposome-encapsulated medication is predicted to be delivered without accelerated deterioration and results in fewer adverse consequences and exhibit greater evidence of resilience in the receiver. Matabudul et al. [121] investigated whether various intervals of Lipocurc intravenous infusions affected metabolism and tissue delivery of curcumin, and whether reacting necropsied beagle dog tissues with phosphoric acid before evaluating curcumin and its active ingredient (tetrahydrocurcumin) could regulate the substances and allow effective assessments. Figure 4 illustrated various nano drug delivery system for curcumin delivery.

Curcumin-loaded nanoparticles have been used to assess liposomal curcumin’s potential against various cancers [122,123,124]. Liposomal curcumin demonstrated significant anticancer potential against osteosarcoma and breast cancer cell lines in vitro and in vivo via the caspase cascade, which resulted in apoptotic cell death. The xenograft osteosarcoma model in vivo was used to demonstrate the efficacy of liposomal curcumin nanoparticles. Curcumin was incorporated into a liposomal delivery system for intravenous administration by Li et al. [125]. These authors also used human pancreatic cancer cells in vitro and in vivo to demonstrate the effects of liposome-encapsulated curcumin on proliferation, apoptosis, signaling, and angiogenesis [126,127]. Curcumin encapsulated in liposomes inhibited tumor angiogenesis in vivo and suppressed pancreatic carcinoma growth in murine xenograft models [128]. It also inhibited proliferation, caused apoptosis, and inhibited the NF-κB pathway in human pancreatic cells in vitro, as well as having anticancer and antiangiogenesis actions in vivo [129,130]. Curcumin partitioning into liposomes containing dimyristoyl phosphatidyl choline (DMPC) and cholesterol suppressed cellular proliferation in human prostate cancer cell lines (LNCaP and C4B2) by 70–80% without affecting viability. When compared to curcumin, oral doses of liposome-encapsulated curcumin resulted in high bioavailability and quicker and better absorption of curcumin in rats [131]. Curcumin is incorporated into liposomes and then reaches cells are demonstrated in Figure 5.

### 5.2. Micelles

A micelle is a spherical vesicle made up of amphiphilic surfactant molecules that spontaneously assemble in water. Micelles are commonly utilized to deliver drugs that are not very water soluble, such as curcumin [133]. Curcumin can be added pre- or post-micelle production to solubilize it within the hydrophobic center of micelles. Micelles are colloidal dispersions that are highly thermally stable and include particulates. Because the particles are so tiny, they do not scatter light waves substantially and thus tend to be optically transparent. Micelle viscosities are determined by surfactant concentration and micelle structure. Micelles often seem to be fluids with low viscosities at low concentrations, but semisolids with greater viscosities or gel-like textures at higher concentrations. Since these variables impact the size, shape, interactions, and dynamics of the colloidal structures produced, the rheological characteristics of micellar systems are highly influenced by the surfactant type and ambient circumstances [134]. Micelles improve the bioavailability of hydrophobic substances (such as curcumin) by enhancing their bioaccessibility in gastrointestinal fluids and potentially boosting epithelial cell penetration [135]. Curcumin encapsulated polymeric micelles (CUR-M) were produced using a one-step solid dispersion method, and the efficacy of CUR-M was tested in a breast tumor model, by Liu et al. [136]. CUR-M exhibited better activity than pure curcumin at suppressing the generation of breast cancers and spontaneous pulmonary metastases of the lungs. Curcumin-poly (ethylene glycol) methyl ether (MPEG-PCL) micelles with solid dispersion improved curcumin’s antiangiogenesis and antitumor effects.

Curcumin micelles may be beneficial in the treatment of lung cancer, according to the findings in [137]. Chang et al. [138] investigated the cell uptake, intracellular localization, and cytotoxicity of different sizes of curcumin encapsulated micelles on human colon cancer cells in vitro. Their findings showed that smaller curcumin-loaded micelles had a greater possibility for inducing cytotoxicity in human colon cancer cells than larger micelles. As a result, drug loading, micelle size, and uptake/release kinetics are all critical factors to consider when it comes to nanoparticle drug delivery [139].

Another study by W. Ma et al. [140] on co-assembly of doxorubicin and curcumin targeted micelles for synergistic delivery by ultrasonication. The micelles demonstrated improved antitumor efficacy, improved tumor suppression and diminution, and improved cytotoxic effects. A study by Yang et al. [141] on a pH multistage responsive micellar system with charge-switch and PEG layer detachment for codelivery of paclitaxel and curcumin to synergistically eliminate breast cancer stem cells using the dialysis method found tumor growth inhibition with no substantial recurrence.

### 5.3. Nanoemulsions

Nanoemulsions (NEs) are biphasic dispersions, either oil-in-water (*o*/*w*) or water-in-oil (w/o), in which particles from one phase are dispersed in another and stabilised by an emulsifier that lowers the interfacial tension between the two immiscible liquids [142]. Regardless of the lack of an existing arrangement on the maximum limit, NEs can have a mean droplet diameter of 20 to 200 nm [143]. NEs can be made with a variety of emulsifiers, including biopolymers (such as proteins and polysaccharides), phospholipids, surfactants (such as Tween 80), and oils (such as fats and essential oils) [144,145,146]. Because of their prospective capacity to encapsulate bioactive chemicals for a variety of applications in the dietary, pharmaceutical, and medical fields, NEs have garnered much attention [147]. Among different nanoformulations, NEs are best suited for lipophilic drugs such as curcumin to improve their solubility and bioavailability [148]. Furthermore, NEs are more resistant to droplet agglomeration and gravitational dispersion than traditional emulsions [149]. As a result, numerous experiments have been conducted to demonstrate the feasibility of utilizing NEs to develop curcumin delivery systems [149]. Thermodynamically stable and transparent NEs are required. Despite this, because NEs are nonequilibrium systems, they take a great deal of energy to develop. Curcumin can therefore be enclosed in NEs with either high- or low-energy techniques. Low-energy techniques depend on the nature (i.e., molecular structure and solubility) of the molecules present in the solution to produce oil droplets by controlling interfacial alteration at the border between two immiscible phases [149].

A few in vitro and in vivo studies have been carried out to evaluate the therapeutic properties (e.g., anticarcinogenicity), biological effectiveness, and patient compliance of NEs containing curcumin. Curcumin enclosed in NEs also had better anticancer impact than free curcumin, according to research. For example, Nikolic et al. [150] studied the pharmacological safety and compliance profile of low-energy NEs stabilized by polysorbate 80 and soybean lecithin as curcumin delivery methods. Curcumin-NEshada PDI of 0.156 to 0.175, with an average size diameter of 150 nm. Curcumin-NEs were found to have considerable a cytotoxic effect against the HeLa cell line (IC50 = 22.89 g/mL) and human melanoma (Fem-x cell line, IC50 = 37.87 g/mL) in cytotoxicity assays. Free curcumin, on the contrary, had IC50 values of 7.77 and 20.64 g/mL, respectively, for HeLa and Fem-x cell lines. The scientists also noted that encapsulation improved curcumin-NEs’ safety profile when compared to free curcumin, with IC50 values of 67.72 and 26.97 g/mL for curcumin-NE and free curcumin, respectively, against a normal lung fibroblast (MRC-5) cell line.

The curcumin-NEs had an average, PDI, and zeta potential of 195 to 217 nm, 0.200, and −30 to −36 mV, respectively. In the Northeast, the efficiency of encapsulation (% EE) and drug loading (% DL) of curcumin were 95 and 2.1%, respectively. Human gastric adenocarcinoma (AGS), colorectal adenocarcinoma (HT29-ATCC), cells generated from HT29-ATCC with enhanced metastatic potential (HT29-US), human mammary gland adenocarcinoma (MDA-MB-231), and murine melanoma cell viability was evaluated after treatment with CUR-NE (B16F10). The IC50 values for AGS, MDA-MB-231, HT29-US, and HT29-ATCC cells were 24, 26.2, 75.7, and 84.6 M, respectively, indicating that this nanoformulation inhibited cancer cell growth [151].

### 5.4. Solid Lipid Nanoparticles

Solid lipid nanoparticles (SLNs) constitute a unique possible colloidal carrier system and comprise physiologically tolerable lipid components. At room temperature, their form is solid. SLNs are a popular way to improve the oral bioavailability of drugs that are not very water soluble [152]. Curcumin was incorporated into SLNs by Kakkar et al. [153] to enhance its oral bioavailability. A microemulsification method was used to generate curcumin-SLNs with a mean particle size of 134.6 nm and a maximum amount of drugs <92%. After oral delivery of curcumin-SLNs, in vivo pharmacokinetics was measured in rat plasma via a validated LC–MS/MS technique. When compared to curcumin-solid lipids, the results showed a substantial improvement in bioavailability times following delivery of curcumin-SLNs. Data have shown that improved and consistent bioavailability can aid in determining a drug’s therapeutic impact. In addition, Kakkar et al. [154] integrated curcumin with SLNs to enhance bioavailability. Plasma and brain cryosections were then examined under a fluorescent/confocal microscope for fluorescence. The biodistribution of 99m Tc-labeled curcumin-SLNs and curcumin-solid lipids in mice following oral and intravenous treatment was also studied. Yadav et al. [155] introduced a new formulation strategy for treating investigational colitis in rats using a colon-specific delivery system. Palmitic acid, stearic acid, and soya lecithin were used to make solid lipid microparticles (SLMPs) containing curcumin with an optimum proportion of poloxamer 188. The colonic delivery mechanism of the SLMP preparation of curcumin was then researched for its antiangiogenic and anti-inflammatory properties by employing chick embryo and rat colitis studies. In the chorioallantoic membrane test, solid lipid microparticles of curcumin were revealed to constitute a strong angioinhibitory drug. When compared to dextran sulfate solution-treated control rats, animals fed with curcumin and its SLMP complex gained weight quicker. When compared to free curcumin and controlled animals, the increase in total colon length was revealed to be substantially higher in solid lipid microparticle-treated rats. Furthermore, rats treated with curcumin-SLMPs had fewer mast cells in their colon mucosa. Colonic distribution of curcumin-SLMPs considerably reduced the severity of colitis produced by dextran sulfate solution administration [155].

In one study on SLNs loaded with curcumin and doxorubicin prepared by a cold dilution microemulsion method, it was found that curcumin-loaded SLNs were more effective than free curcumin at increasing doxorubicin efficacy in resistant TNBC cells. The curcumin-loaded SLNs decreased P-gp activity and expression and decreased P-gp transcription by reducing intracellular ROS and NF-κB activity. They also restored doxorubicin sensitivity by downregulating the ROS/NF-κB/Pgp axis in resistant TNBC cells cocultured with macrophages [156].

Another study showed that curcumin-SLNs prepared using an emulsification evaporation–low temperature solidification method resulted in drug loading and encapsulation efficienciesof23.38% and 72.47% and greater cytotoxicity against SKBR3 cells. In an in vitro cellular uptake study, curcumin-SLNs were found to have a good absorption efficiency by SKBR3 cells. Curcumin-SLNs also produced higher apoptosis in SKBR3 cells and decreased the manifestation of cyclin D1 and CDK4.These data suggest that curcumin-SLNs might be a promising chemotherapeutic formulation for breast cancer therapy [156].

### 5.5. Niosomes

Niosomes are tiny lamellar structures containing cholesterol and a nonionic surfactant of the alkyl or dialkyl polyglycerol ether family [157]. Because of hydrophilic nature niosomes can serve as a vessel for drug substances with a large variety of solubilities. They have amphiphilic, lipophilic, and amphiphilic moieties in their composition, and they act similarly to liposomes in vivo, making them a viable substitute to liposomal drug carriers. These properties are dependent on the bilayer’s constituent and the method of production [157]. The type of surfactant, the nature of the encapsulated medication, the storage temperature, the detergents, and the usage of bilayer lipids can all impact the stability of niosomes [158]. Niosomes are also expected to have a variety of therapeutic uses, including anticancer, and it can enhance drug therapeutic parameters by limiting drugs’ effect on target cells. They also use a new drug delivery system to enhance the oral bioavailability of improperly absorbed drugs, such as curcumin, and boost drug penetration via the skin [159]. Curcuminoid niosomes were generated with a variety of nonionic surfactants to improve skin penetration of curcuminoids in a report by Rungphanichkul et al. [160]. Entrapment potency and in vitro curcuminoid penetration through snakeskin were used to assess the results. When compared to a vehicle solution of curcuminoids, niosomes significantly improved curcuminoid penetration. Curcuminoids could be successfully produced as niosomes, according to the data, and such preparations offered superior characteristics for transdermal drug delivery [161].

### 5.6. Dendrimers

Dendrimers are polymers with a monodisperse, three-dimensional structure with nanoscale dimensions that are highly branching, synthetic, and radially symmetric [162]. Dendrimers are made up of two parts: (i) a central initiator core made up of a molecule or an atom with at least two similar chemical functions, and (ii) an inner structure made up of branches emanating from the core and constructed by repeat units with at least one branch junction [163]. As a consequence, a radial series of concentric layers known as “generations” is formed, which forms an exterior structure composed of terminal functional groups present on dendrimer surfaces [164]. Dendrimers can encapsulate both hydrophilic and hydrophobic bioactive substances and have been utilized to increase the bioavailability and water solubility of hydrophobic substances [164]. Dendrimers’ solubility may be easily increased by adjusting core, branch, and surface functions or changing the surface with hydrophilic moieties [165].

The immobilisation of gold nanoparticles on folate-conjugated dendritic mesoporous silica-coated reduced graphene oxide nanosheets provided a novel nanoplatform for curcumin distribution that was pH-controlled and targeted [166]. According to Wang et al. [167], incorporating curcumin with a PAMAM dendrimer enhanced curcumin’s water solubility 200-fold over free curcumin. In addition, the PAMAM dendrimer resulted in prolonged curcumin release, increased anticancer efficacy against A549 cell lines, and reduced intracellular ROS production [168].

Dendrimer nanoparticle articulated curcumin was formulated as a therapeutic target for glioblastoma in mice. The anti-inflammatory properties of D-Cys-Cur and D-Cys transfection of GL261 cells were similar. In mice, treatment with both complexes resulted in a similar increase in lifespan. There was no difference in the size of mouse tumors between treatment groups [169]. Accordingly, NP-encapsulated curcumin was found effective in decreasing the growth of medulloblastoma cell lines (i.e., DAOY and D283Med) and GBM neurosphere lines (i.e., HSR-GBM1 and JHH-GBM14), particularly affecting the CD133+ stem-like population [170]. Babaei et al. [171] developed curcumin-loaded dendrosomes and evaluated them in vitro and in a dose- and time-dependent manner. The dendrosomes had greater bioavailability and inhibitory effects against WEHI-164 and A431 cancer cell lines. Henceforth curcumin induced apoptosis.

In another study, curcumin dendrosomes were prepared by Tahmasebi et al. [172] as a nontoxic nanocarrier with chemical and physical stability and a spherical form. The dendrosomes had an inhibitory impact on U87MG glioblastoma cancer cell lines but had no effect on nonneoplastic cells. In another study, new dendritic silica/titania mesoporous nanoparticles (DSTNs) loaded with curcumin were synthesized and coated with polyethylenimine–folic acid groups (PEI–FA). FA groups on the surfaces of DSTNs increased cancer cell absorption through receptor-mediated endocytosis. The quantity of curcumin released from DSTNs could be regulated by tweaking the US radiation time, according to the release profiles of the CUR–PEI–FA–DSTN system. MTT cytotoxicity assays of free curcumin, free PEI–FA–DSTN nanocarrier, and CURL–PEI−FA–DSTNs against A549 (human lung cancer cell lines) and HeLa (human cervical carcinoma cell lines) revealed that CUR–PEI–FA–DSTNs were more harmful than curcumin and PEI–FA–DSTNs alone. The novel system, CUR–PEI–FA–DSTNs, were deemed a powerful drug delivery system for improving the efficacy of curcumin’s anticancer activity in combination chemosonodynamic treatment [173].

### 5.7. Nanogels

Curcumin’s biological activity can be amplified using nanogels. A nanogel is a nanoparticle (10 to 100 nm) made up of a hydrogel created by controlled physical or chemical cross-linking of polymers. Nanogel’s cross-linked structure provides a stable platform for drug storage and release. It is a method for preparing and delivering active medicines to cells in order to maintain activity, improve stability, and prevent drug immunogenicity [174]. Reeves et al. [175] developed and tested a colloidal nanogel carrier method for encapsulating curcuminin order to improve its solubility and cytotoxicity. When compared to curcumin alone, this curcumin–nanogel combination was responsible for destroying tumor cells. Dandekar et al. [176] combined hydroxypropyl methylcellulose and polyvinyl pyrrolidone to make curcumin-loaded hydrogel nanoparticles, which they evaluated for antimalarial potential in mice. It was demonstrated that curcumin-loaded hydrogel nanoparticles had a significant effect [177,178].

Curcumin nanoemulgel showed improved transdermal penetration against squamous cell carcinoma in research. The nanoemulgel produced substantially more drug release with substantially less toxicity [179]. The in vitro cytotoxic effect of curcumin and nanocurcumin on the human breast cancer cell line (MDAMB231) was investigated [180]. The process of self-assembly was used to make meristic acid chitosan (MAchitosan) nanogels. The nanogels were loaded with curcumin. Curcumin-loaded nanogels were shown to be at least twice as powerful as free curcumin, perhaps because of increased absorption [180]. Self-assembled nanogels made from hydrophobically modified dextrin were efficient curcumin nanocarriers, according to a study on the stability and loading effectiveness of curcumin loaded nanogels. This study showed that the preparation was more stable than a preparation in phosphate-buffered saline, as determined by dynamic light scattering and fluorescence tests. The therapeutic efficacy of curcumin-loaded nanogels in HeLa cell cultures was also examined in the paper. The nanogels were just as efficient as free CUR at inhibiting cancer cell proliferation [181].

### 5.8. Cyclodextrins

Cyclodextrins are a type of cyclic oligosaccharide made up of sugar molecules linked together [182]. To make them from starch, an enzyme switch is utilized. Cyclodextrins are glucose cyclic oligomers that can be combined with nanoparticles and fragments of larger complexes to produce water-soluble inclusion complexes. Cyclodextrins are also employed in agriculture and environmental research, drug delivery systems as well as synthetic industry [183].

Tønnesen et al. [184] used the UV/VIS scanned spectrophotometer and HPLC methods to produce cyclodextrin–curcumin complexes to enhance the water solubility and hydrolytic stability of curcumin. The complex improved curcumin’s hydrolytic stability and increased its photodecomposition rate in organic solvents when compared to the untreated curcumin. As a consequence, the stability and degradation of curcumin were impacted by the cavity size and charge of cyclodextrin side chains [185]. Curcumin compounds were shown to be more resistant to hydrolytic decomposition in cyclodextrin solutions than pure curcumin in previous findings on their hydrolyzing and thermo chemical durability, solubility, and phase distribution [186]. Another study used cyclodextrin-based nanosponges to improve the solubility of curcumin. When compared to untreated curcumin and aγ-cyclodextrin complex, the loaded nanosponges had higher solubilization efficiency. Curcumin’s interactions with nanosponges were verified by characterization of the curcumin–nanosponge complex. Curcumin drug release in vitro was also regulated over a long length of time, and the combination was nonhemolytic [187]. Below mentioned Table 2 has characterization properties of various nanoparticle- conjugated curcumin for the treatment of various diseases.

## 6. Clinical Studies and Patent Reviews

Nanocurcumin has been shown to be helpful in malignancy, multiple sclerosis, amyotrophic lateral sclerosis, ankylosing spondylitis, renal failure, and metabolic syndrome patients in many clinical studies. In Table 3, various nanocurcumins in clinical studies are mentioned, and Table 4 illustrated various nanocurcumin formulations in patent reviews.

## 7. Conclusions and Future Perspectives

Curcumin has generated much attention throughout the years because of its potential therapeutic applications. Based on thorough research, nanoencapsulation methods improved the pharmacokinetic characteristics of the curcumin formulation and provided more excellent therapeutic benefits. According to the information covered in the different units of this study, several curcumin nanoformulations have been produced and utilized to treat various diseases in humans, and curcumin nanoformulation has made remarkable development over the last decades. Curcumin, apart from targeting afflicted cells, interferes with healthy cells and tissues; hence, tissue specificity is an area that must be investigated. Other main challenges that must be addressed include storage stability and reducing production costs. However, using curcumin-loaded NPs in combination with the main therapeutic agent allows for a lower dose of the main therapeutic agent, enhancing therapeutic potential while lowering systemic toxicity.

Furthermore, while functionalized nanoparticles provide effective drug targeting, their nanosize structure and huge surface area may cause particle aggregation and low drug loading [230,231]. The toxicity of nanoparticles is affected by their state of aggregation and mechanical properties, which are dependent on their production and purifying methodologies. Thus, more research is needed to develop curcumin-loaded NPs with lesser toxicity. Concerns about the toxicity of NP-based delivery methods include neuroinflammation, excitotoxicity, and allergic reactions [232]. These can be addressed via a complete investigation of the chemicals used in nanocurcumin encapsulation, ensuring minimum cytotoxicity and increased biocompatibility. Furthermore, the majority of studies to date have focused on nanocurcumin’s in vitro effects. A series of conclusive in vivo tests in various disease experimental models are required to provide a more definite platform for promoting nanocurcumin up to the level of clinical trials.

## Figures and Tables

**Figure 1 molecules-26-07109-f001:**
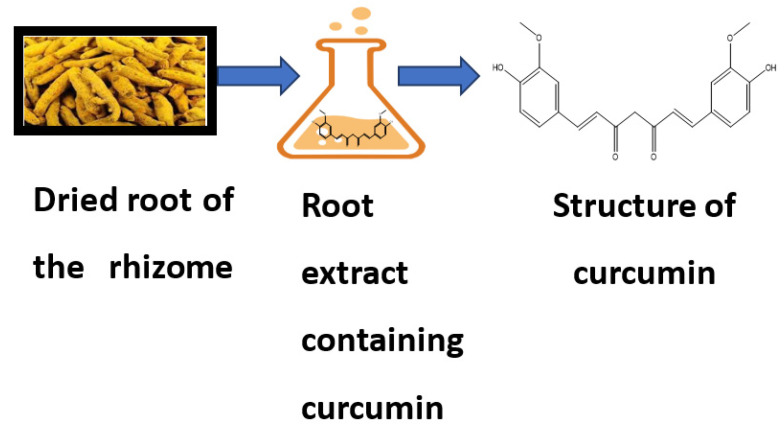
Source and chemical structure of curcumin.

**Figure 2 molecules-26-07109-f002:**
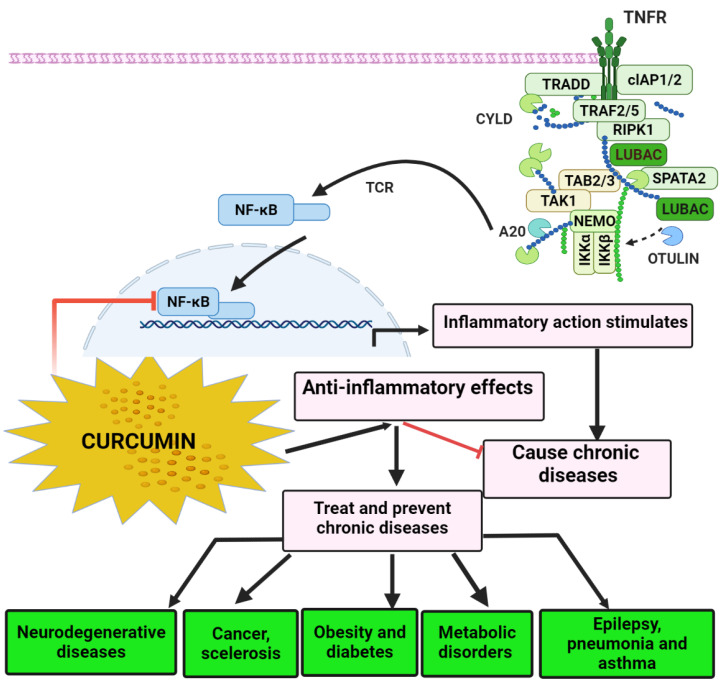
Inflammatory action leading to chronic diseases, and mechanism of action of curcumin against inflammatory response.

**Figure 3 molecules-26-07109-f003:**
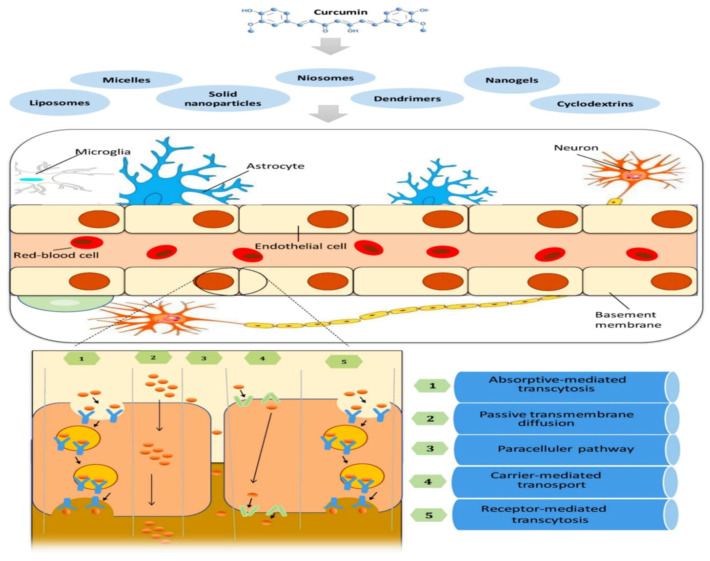
The major types of curcumin nanocarriers, the structure of the blood–brain barrier (BBB), and the BBB crossing mechanism.

**Figure 4 molecules-26-07109-f004:**
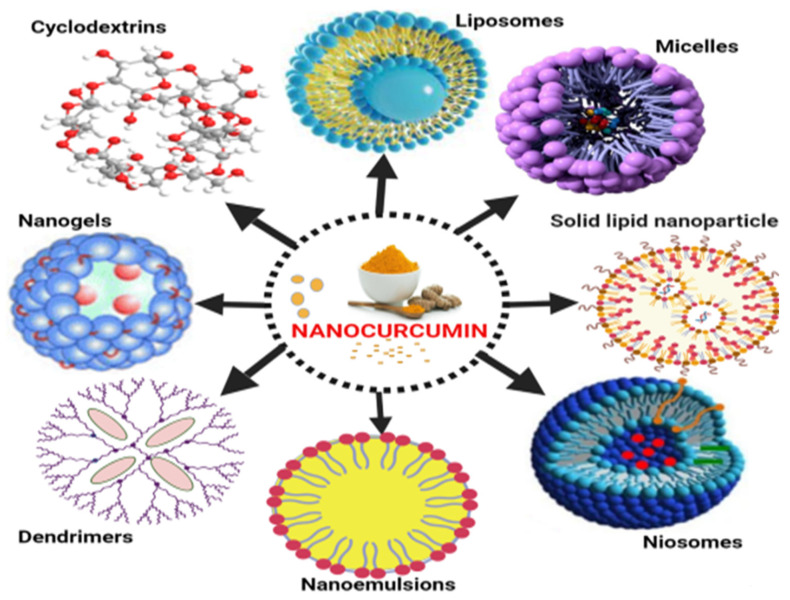
Various nano drug delivery systems for curcumin.

**Figure 5 molecules-26-07109-f005:**
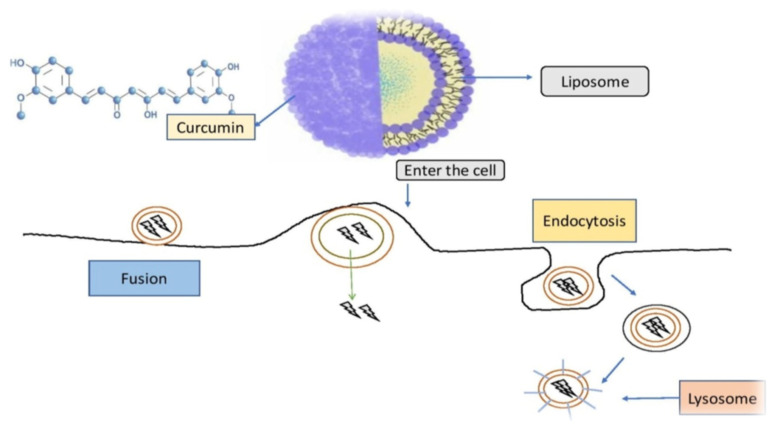
Curcumin is incorporated into liposomes and then reaches cells in this diagram. Curcumin is enclosed within the liposomal vessel and chemically attached to the liposome, preventing it from being destroyed on its approach to the target. Phospholipids, which are found in biological membranes and may deliver curcumin to cells via fusion and endocytosis, are frequently used in liposome membranes [132].

**Table 1 molecules-26-07109-t001:** The effects of curcumin on mechanisms involved in the degeneration in AD.

Mechanisms Involved in Degeneration in AD	Effects of Curcumin
β-amyloid	
Increased production	Decrease in β-amyloid
β-sheet formation	Inhibition of sheet formation
Neurotoxicity	Decrease in neuronal toxicity
NF-κB activation	Decrease in NF-κB activation
ERK1/2	Decrease in ERK-1/2 expression
γ-secretase activity	Inhibition of γ-secretase
Oxidative stress	
IL-1β	Decrease in IL-1β
GSK-3β	Decrease in GSK-3β
Caspase-3	Prevention ofβ-amyloid-induced damage
Akt	Activate neuroprotective pathway

Abbreviations: NF-κB, nuclear factor kappa B; ERK1/2, extracellular signal regulated protein kinase; IL-1β, interleukin 1 beta; GSK-3β, glycogen synthase kinase-3beta; Akt, protein kinase B.

**Table 2 molecules-26-07109-t002:** Characterization of nanoparticle-conjugated curcumin for the treatment of various diseases.

Types	Form	Size (nm)	Study Models	Models	Outcomes	References
Liposome	Globular	25–205	Breast cancer, lung cancer, renal ischemia, and malaria	In vivo, in vitro	Antitumor and antiangiogenesis effects were improved; antimelanoma, anti-inflammatory, and antimalarial activities were demonstrated	[188,189,190]
Micelle	Spherical	10–100	Lung, colorectal, and breast cancer	In vivo, in vitro	Improved antioxidative and anticancer properties; enhanced solubility and bioavailability; longer circulation period and enhanced fluorescence impact	[191,192,193,194,195,196,197]
Solid lipid nanoparticles	Spherical	50–1000	Ischemia of the brain, colitis, allergies, and breast cancer	In vivo, in vitro	Improved anti-inflammatory properties; enhanced blood circulation; enhanced brain delivery	[198,199,200,201,202]
Niosome	Lamellar	190–1140	Cancer cells	In vivo, in vitro	Improved fluorescence intensity; anticancer properties	[158]
Dendrimer	Globular polymer	15–150	Breast cancer and colon cancer	In vivo, in vitro	Enhanced antitumor and antiproliferative effects; improved stability	[203,204,205,206,207,208]
Nanogel	Network of cross-linked polymers	10–200	Melanoma and breast, pancreatic, colorectal, andskin cancer	In vitro	Improved fluorescence effects; improved bioavailability; increased anticancer activity; more regulated release; extended half-life; improved melanoma therapy	[209,210,211,212,213]
Cyclodextrin	Cyclic	150–500	Cancers of the bowel, breast, lung, pancreas, and prostate	In vivo, in vitro	Increased solubility; stronger antiproliferation effects; improved anticancer and anti-inflammatory properties; Improved bioavailability	[214,215]

**Table 3 molecules-26-07109-t003:** Nanocurcumin in clinical trial studies.

S. No.	Clinical Studies Identifier	Study Title	Interventions	Phase, Recruitment Status	Place Intended for Study	References
1	IRCT2017080135444N1	The synergistic effects of nanocurcumin and coenzyme Q10 supplementation in migraine prophylaxis: a randomized, placebo-controlled, double-blind trial	Migraine-related impairment	Phase 2 and 3, completed	Tehran University of Medical Sciences, Tehran, Iran	[216]
2	NCT02532023	Effects of nanocurcumin on inflammatory factors and clinical outcomes in critically ill patients with sepsis: A pilot randomized clinical trial	Patients with sepsis who are severely sick	Phase 4, completed	Tabriz University of Medical Sciences, Tabriz, Iran	[217]
3	IRCT20200705048018N1	The effects of nanocurcumin as a nutritional strategy on clinical and inflammatory factors in children with cystic fibrosis: the study protocol for a randomized controlled trial	Cystic fibrosis	Phase 1, Recruiting	Akbar Children’s Hospital, Mashhad, Iran.	[218]
4	IRCT20161208031300N1	Impact of resistance exercises and nanocurcumin on synovial levels of collagenase and nitric oxide in women with knee osteoarthritis	Osteoarthritis	Phase 3, completed	Imam Ali Hospital, Bojnourd, Iran	[219]
5	IRCT20190523043678N1	Combination Therapy with 1% Nanocurcumin Gel and 0.1% Triamcinolone Acetonide Mouth Rinse for Oral Lichen Planus: A Randomized Double-Blind Placebo Controlled Clinical Trial	Oral lichen planus	Phase 3, completed	Shahid Beheshti University of Medical Sciences, Iran	[220]
6	NCT03150966	The immunomodulatory effects of oral nanocurcumin in multiple sclerosis patients	Multiple sclerosis	Phase 2, completed	Tabriz University of Medical Sciences, Iran	[221]
7	NCT01201694	Study on surface controlled water soluble curcumin	Cancer	Phase 1, completed	UT MD Anderson Cancer Center Houston, Texas, United States	[222]

**Table 4 molecules-26-07109-t004:** Nanocurcumin in patent reviews.

S. No.	Patent No.	Study Title	Interventions	References
1	US 9, 931, 309 B2	Complex curcumine-sophorolipids	Nanoencapsulated in acidic sophorolipids to enhance curcumin’s bioavailability and solubility to boost its pharmacological response, including cancer	[223]
2	US20180028447	Development of curcumin and piperine loaded double-layered biopolymer based nano delivery systems by using electrospray/coating method	Curcumin was contained in the core network, which was made up of zein protein, and piperinewas encased in the outermost casing, which was chitosan. Although the precise method emphasizing the molecular mechanism of piperine for curcumin improvement was not defined, it was demonstrated that decreasing the efficiency of cytochrome P4503A4 (CYP3A4), which plays a role in curcumin metabolism, enhanced the residence duration of curcumin.	[224]
3	US20100290982A1	Solid in oil/water emulsion-diffusion-evaporation formulation for preparing curcumin-loaded PLGA nanoparticles	Findings were obtained by producing the solid in oil/water emulsion diffusion evaporation composition for generating curcumin-loaded PLGA nanoparticles.	[225]
4	US10413511B2	Liposomal formulations of lipophilic compounds	Revealed unique preparations for curing refractory and resistant pancreatic malignancies with a paclitaxel in a cationic liposomal form; gemcitabine, a ribonucleotide reductase inhibitor that prevents DNA synthesis in cancerous cells; and other anticancer drugs	[226]
5	US9138411B2	Curcumin-ER, a liposomal-PLGA sustained release nanocurcumin for minimizing QT prolongation for cancer therapy	The bioactive substance curcumin and curcumin–PLGA analogues were utilized in the liposome, which consisted of a polymeric core with ground lipidic components. Human embryonic kidney (HEK 293) cell lines treated with the human ether-related gene (hERG) were used to test it. The whole-cell patch-clamp present review and approval approach was used to examine the in vitro consequences of the curcumin liposomal preparation of potassium-selective IKr currents produced in normoxia in stably transfected HEK 293 cells.	[227]
6	US9283185B2	Liposomal curcumin for treatment of cancer	In human patients, curcumin analogues and curcumin enclosed as liposomal preparations were revealed to treat pancreatic cancer, breast cancer, and melanoma.	[228]
7	WO 2013132457	Nanocrystalline solid dispersion compositions and process of preparation thereof	A curcumin–stearic acid mixture was produced and nebulized by spray-drying to create a dry powder to produce nanocrystalline solid dispersion. Oral dosing of spray-dried curcumin and curcumin–stearic acid nanocrystalline solid dispersion in rats resulted in a 15-fold increase in curcumin oral bioavailability with nanocrystalline solid dispersion compared to the control.	[229]

## Data Availability

Not Applicable.

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
