# Peer review of "The Multifaceted Role of Curcumin in Advanced Nanocurcumin Form in the Treatment and Management of Chronic Disorders"

_molecules, 2021, doi:10.3390/molecules26237109_

Round 1
Reviewer 1 Report
Thanks for providing this informative review on Curcumin delivery. In my book, there are several reviews on this topic that were previously published, so it was much better if you could opt for a special topic in this regard such as the effect of curcumin on a special disorder or curcumin delivery to the special organ. However, I liked the updated references you have used in this work. Most of them are completely useful and inspiring and it would a plus point for your manuscript. Several comments must be considered before publishing this review.
Please do apply these changes.
- Please check the plagiarism and duplication of the text and decrease it as much as you can. Our plagiarism checker report shows that some sentences are a copy of their original text!
- The English of the text needs improvement. Some grammatical errors in some parts of the review can be found. Please double-check it.
- What does Figure 1 show? Where is the caption? Please replace it with a creative one and write its caption as well.
- Some repetitive explanations of curcumin and the nanocarriers can be seen in the whole text. You should remove them. Please prevent mentioning some clear concepts about curcumin since they have been discussed myriads of time in the original papers, reviews, chapters, and so forth.
For instance, some initial sentences of the introduction, and general information about the structure of each nanoparticle in each paragraph are repeated and boring. You must focus on the main subject to motivate readers for following your review with state-of-the-art discoveries regarding the curcumin applications for disorder treatments!
- The authors can also use the following references:
- Zhang, H., van Os, W.L., Tian, X., Zu, G., Ribovski, L., Bron, R., Bussmann, J., Kros, A., Liu, Y. and Zuhorn, I.S., 2021. Development of curcumin-loaded zein nanoparticles for transport across the blood-brain barrier and inhibition of glioblastoma cell growth. Biomaterials Science.
- Malekmohammadi, S., Hadadzadeh, H., Farrokhpour, H. and Amirghofran, Z., 2018. Immobilization of gold nanoparticles on folate-conjugated dendritic mesoporous silica-coated reduced graphene oxide nanosheets: a new nanoplatform for curcumin pH-controlled and targeted delivery. Soft matter, 14(12), pp.2400-2410.
- Malekmohammadi, S., Hadadzadeh, H., Rezakhani, S. and Amirghofran, Z., 2019. Design and synthesis of gatekeeper coated dendritic silica/titania mesoporous nanoparticles with sustained and controlled drug release properties for targeted synergetic chemo-sonodynamic therapy. ACS Biomaterials Science & Engineering, 5(9), pp.4405-4415.
Author Response
Cover Letter to the Editor’s and Reviewer’s Comments on Review Manuscript
Manuscript Title: The Multifaceted role of curcumin in advance nanocurcumin form in the treatment and management of chronic disorders
As per reviewer’s comments, we made the following revision.
I am returning here with the above manuscript newly revised.
- Necessary modifications has been made on the manuscript following reviewers comments.
- According to reviewer’s comment everything has been corrected by authors and with the help of reviewers in the newly revised manuscript.
Appended to this letter is our point-by-point response to the comments raised by the editor’s. We would like to take this opportunity to express our sincere gratitude to the editor for the insightful comment. We would also like to thank you for allowing us to resubmit a revised copy of the manuscript.
Now we are anticipating this revised manuscript will be more suitable on your journal.
Cordially yours,
Mr. Rahman
S.No. |
Reviewer I Comment |
CORRECTED MANUSCRIPT TEXT (All corrections are highlighted by red color in manuscript text and for major deletion or addition used track change) |
1 |
Please check the plagiarism and duplication of the text and decrease it as much as you can. Our plagiarism checker report shows that some sentences are a copy of their original text!
|
Thanks for your valuable comments. Authors removed the repetition in the sentences and rephrased many sentence and highlighted by red color in manuscript. Line no. 109-114: rephrased Line no. 136-140: rephrased Line no.144: revised Line no.160-161: revised Line no.191-193: rephrased Line no.449-452: rephrased Line no. 458-463: added new text Line no.557-567: added new text Line no.658-660: revised
|
2 |
The English of the text needs improvement. Some grammatical errors in some parts of the review can be found. Please double-check it.
|
Thanks for your valuable suggestion. Authors have corrected the grammatical error as well as English this throughout the manucript also highlighted by red color in manuscript.
|
3 |
What does Figure 1 show? Where is the caption? Please replace it with a creative one and write its caption as well. |
Thanks for your valuable comments. Authors has mentioned the caption of figure 1 and highlighted the sentence by red color in manuscript. Line no. 94: Figure 1. Source and chemical structure of Curcumin
|
4 |
Some repetitive explanations of curcumin and the nanocarriers can be seen in the whole text. You should remove them. Please prevent mentioning some clear concepts about curcumin since they have been discussed myriads of time in the original papers, reviews, chapters, and so forth.
|
Thanks for your valuable comments and suggestion. Authors have removed all repetitive explanations throughout the manuscript also mentioned through track changes and highlighted the sentence by red color in manuscript. Removed lines no with correction by tract change. 120, 130, 131134, 140-143, 148-152,161-165166-169,309,312,328-329,356-358,369-371,463-466,507-508,586-587,652-653,673-694,702-705,723,974,983-984. |
5 |
The authors can also use the following references:
|
Thanks for your valuable suggestion on references. Authors have added the studies from this article in the manuscript and references also added in references section and highlighted the sentence by red color in manuscript. Line no. 579-586, Reference 86: Another research in which curcumin-loaded zein nanoparticles (CUR-ZpD-G23 NPs) successfully crossed the BBB and administered curcumin to glioblastoma cells. When compared to free curcumin, the NPs increased cellular absorption of curcumin by C6 glioma cells and demonstrated great penetration into 3D tumour spheroids. BBB crossing and tumour spheroid penetration were increased by G23-functionalized NPs. Furthermore, in liquid and soft agar models of C6 glioma cell development, the NPs significantly reduced proliferation and migration, as well as causing cell death[86] Line no. 9619-971, Reference 166: Immobilization of gold nanoparticles on folate-conjugated dendritic mesoporous silica-coated reduced graphene oxide nanosheets: a new nanoplatform for CUR pH-controlled and targeted delivery [166]. Line no. 992-1004, Reference 173: In another study, new dendritic silica/titania mesoporous nanoparticles (DSTNs) loaded with curcumin were synthesized and coated with polyethylenimine-folic acid groups (PEI-FA). FA groups on the surfaces of DSTNs increase cancer cell absorption through receptor-mediated endocytosis, according to the findings. The quantity of curcumin released from DSTNs is regulated by tweaking the US radiation time, according to the release profiles of the CUR-PEI-FA-DSTN system. MTT cytotoxicity assays of free curcumin, free PEI-FA-DSTN nanocarrier, and CUR-PEI-FA-DSTNs against A549 (human lung cancer cell lines) and HeLa (human cervical carcinoma cell lines) revealed that CUR-PEI-FA-DSTNs are more harmful than curcumin and PEI-FA-DSTNs alone. According to these findings, the novel system, CUR-PEI-FA-DSTNs, may be deemed a powerful drug delivery system for improving the efficacy of curcumin's anticancer activity in combination chemo-sonodynamic treatment [173]. |
Reviewer 2 Report
This article provides a detailed review, especially of curcumin-nano drug delivery.
However, I think the following improvements are needed.
The anti-inflammation mechanism is too simplified, and curcumin has recently been shown to affect upstream anti-inflammatory genes such as, CDGSH iron-sulfur domain 2.
It would be preferable to also clarify the relationship between inflammation and mitochondrial dysfunction.
Regarding the descriptions in 4.1-4.8, it is often confusing whether it is curcumin or nano-curcumin? The authors should revise it significantly
The patent reviews section looks too disorganized and should be structured systematically, e.g., in a table.
Finally, this broad topic lacks future perspectives.
Author Response
Thank you for your letter dated 01 Nov 2021. We are pleased to know that we can revise and resubmit our manuscript.
As reviewer’s comments, we made the following revision.
I am returning here with the above manuscript newly revised.
- Necessary modifications has been made on the manuscript following reviewers comments.
- According to reviewer’s comment everything has been corrected by authors and with the help of reviewers in the newly revised manuscript.
S.No. |
Reviewer II |
CORRECTED MANUSCRIPT TEXT (All corrections are highlighted by red color in manuscript text and for major deletion or addition used track change also) |
1 |
This article provides a detailed review, especially of curcumin-nano drug delivery. However i think the following improvements are needed. The anti-inflammation mechanism is too simplified, and curcumin has recently been shown to affect upstream anti-inflammatory genes such as, CDGSH iron-sulfur domain 2 and it would be preferable to also clarify the relationship between inflammation and mitochondrial dysfunction.
|
Thanks for your valuable comments and suggestion. Authors have added this effect and relationship between inflammation and mitochondrial dysfunction under section 4.2 Anti-Inflammatory Effects and highlighted the sentence by red color in manuscript. Line No.:213-228 Excessive inflammatory activation for long periods of time can result in mitochondrial dysfunction. Traumatic brain injuries (TBI), spinal cord injuries (SCI), and hemorrhagic/ischemic stroke all causes substantial changes in mitochondrial dynamics, notably increased membrane permeabilization, oxidative phosphorylation, and the accumulation of mitochondrial ROS [31,32].Excessive glial activation has been linked to the same outcomes as long-term inflammation. Advanced mitochondrial dysfunction has also been demonstrated to increase inflammatory processes, leading in neuronal injury and poor neurological consequences [33,34].Studies have identified that the gene CDGSH iron sulfur domain 2 (CISD2) protects against inflammatory reactions and apoptosis caused by mito-chondrial dysfunction. The presence of CISD2 in the outer membrane of mitochondria has also been linked to the preservation of mitochondrial integrity. CISD2 deficiency resulted in mitochondrial dysfunction and cell death, according to reports on CISD2 knockout mice [35, 36].The attachment of BCL2 to BECN1, which governs cellular autophagy/apoptosis, has been found to be aided by CISD2. Anti-inflammatory and/or anti-apoptotic therapeutics based on CISD2 are extremely likely to be developed to combat the consequences of aging, neurodegenerative disease, and CNS trauma [37].
|
2 |
Regarding the descriptions in 4.1-4.8, it is often confusing whether it is curcumin or nano-curcumin? The authors should revise it significantly
|
Thanks for your valuable comments. Authors have revised the whole section from 4.1 – 4.8 and illustrated the nanocurcumin effects and also removed the confused the sentences. Highlighted the sentence by red color in manuscript also used track change for corrections where suitable. Revised Line No.:191-193, 458-466, 557-567,579-586 Deleted line :673-694, 702-705 by track change
|
3 |
The patent reviews section looks too disorganized and should be structured systematically, e.g., in a table.
|
Authors are highly obliged for your valuable remark and suggestions on this. Authors revised the Table 3 and added new Table 4 to organize the patent reviews section. Highlighted the sentence by red color in manuscript. Line no. 1058-1065
|
4 |
Finally, this broad topic lacks future perspectives.
|
Authors are highly obliged for your valuable remark and suggestions on this. Authors rewrite section 7. Conclusions and future perspectives and highlighted the corrections by red color in manuscript. Line no. 1074-1091 7. Conclusions and future perspectives Curcumin has generated a lot of attention throughout the years because of its potential therapeutic applications. Based on thorough research, nanoencapsulation methods improved the pharmacokinetic characteristics of the curcumin formulation and provided more excellent therapeutic benefits. According to the information covered in the different units of this study, several curcumin nanoformulations have been produced and utilized to treat various diseases in humans, and curcumin nanoformulation has made remarkable development over the last decades. Curcumin, apart from targeting the afflicted cells, interferes with healthy cells and tissues, hence ensuring tissue specificity are an area that must investigate. . Other main challenges that must address include storage stability as well as reducing production costs. However, using curcumin-loaded NPs in combination with the main therapeutic agent allows for a lower dose of the main therapeutic agent, enhancing therapeutic potential while lowering systemic toxicity. Furthermore, while functionalized nanoparticles provide effective drug targeting, their nano-size structure and huge surface area may cause particle aggregation and low drug loading [232,233]. The toxicity of nanoparticles is affected by their state of aggregation and mechanical properties, dependent on their production and purifying methodologies. Thus, more research is needed to develop curcumin-loaded NPs with lesser toxicity. Concerns about the toxicity of NPs-based delivery methods include neuroinflammation, excitotoxicity, and allergic reactions [234]. This can be accomplished by doing a complete investigation of the chemicals used in nanocurcumin encapsulation, ensuring minimum cytotoxicity and increased biocompatibility. Furthermore, the majority of studies to date have focused on nanocurcumin's in vitro effects. A series of conclusive in vivo tests in various disease experimental models are required to provide a more definite platform for promoting nanocurcumin up to the level of clinical trials.
|
Round 2
Reviewer 2 Report
All the points raised have been addressed.